# Transcript Abundance Patterns of 9- and 13-Lipoxygenase Subfamily Gene Members in Response to Abiotic Stresses (Heat, Cold, Drought or Salt) in Tomato (*Solanum lycopersicum* L.) Highlights Member-Specific Dynamics Relevant to Each Stress

**DOI:** 10.3390/genes10090683

**Published:** 2019-09-05

**Authors:** Rakesh K. Upadhyay, Avtar K. Handa, Autar K. Mattoo

**Affiliations:** 1Sustainable Agricultural Systems Laboratory, USDA-ARS, Henry A. Wallace Beltsville Agricultural Research Center, Beltsville, MD 20705-2350, USA; 2Department of Horticulture and Landscape Architecture, Purdue University, W. Lafayette, IN 47907-2010, USA

**Keywords:** *Solanum lycopersicum*, *LOX*, lipoxygenase, methyl jasmonate, hormone, gene expression, abiotic stresses

## Abstract

Lipoxygenases (LOXs; EC 1.13.11.12) catalyze the oxygenation of fatty acids to produce oxylipins including the jasmonate family of plant hormones. The involvement of jasmonates in plant growth and development and during abiotic stress has been documented, however, the response and regulation of each member of the *LOX* gene family under various abiotic stresses is yet to be fully deciphered. Previously, we identified fourteen members of the tomato *LOX* gene family, which were divisible into nine genes representing the 9-LOX family members and five others representing the 13-LOX family members based on the carbon oxidation position specificity of polyunsaturated fatty acids. Here, we have determined the transcript abundance patterns of all the 14 *LOX* genes in response to four independent abiotic stresses, namely, heat, cold, drought and salt. Our results show that each of these stresses leads to a time-dependent, variable or indifferent response of specific and different set(s) of LOX gene members of both subfamilies, differentiating functional relevance of the 14 *LOX* genes analyzed. Out of the 14 gene members, three *LOX* genes were expressed constitutively or were non-responsive to either heat (*SlLOX9*), cold (*SlLOX9*) or salt (*SlLOX4*) stress. An in-silico *LOX* gene promoter search for stress-responsive elements revealed that only some but not all of the *LOX* genes indeed are decorated with specific and known stress responsive cis-acting elements. Thus, these data implicate some other, yet to be discovered, cis-acting elements present in the *LOX* gene family members, which seemingly regulate tomato responses to defined abiotic stresses presented here.

## 1. Introduction

Plants being sessile are constantly exposed to changes in the environment. Such exposure to different abiotic stresses—heat, cold, drought or salinity—affects photosynthesis, crop yield, nutrition and fruit set leading to reduced crop productivity. Some plants are better than others in developing mechanisms to counterbalance environmental stress conditions. Plant responses to environmental changes have been documented in some species and clearly complex endogenous networking is apparent involving, among other responders, many unique molecules including hormones, transcriptional factors and unique genes [1,2]. Among the hormones, the jasmonate (JA) family members are considered as plant stress hormones since they have been shown to activate plant defense against environmental stresses, including cold, high temperatures, drought and salinity [3,4,5,6,7,8,9,10]. Jasmonates are a class of oxylipins derived from fatty acid hydroperoxides and involve the enzyme lipoxygenase (linoleate: Oxygen oxidoreductase, EC 1.13.11.12 [11,12].

LOX is a family of non-heme-iron-containing fatty acid dioxygenases widely distributed in both plants and animals [13]. LOXs are broadly classified into two sub-families, 9-LOX and 13-LOX, based on the preferred addition of molecular oxygen either at carbon atom 9 or 13 of the hydrocarbon backbone, respectively [7]. JA biosynthesis is catalyzed by 13-LOXs via 13(S)-hydroperoxy-octadecatrienoic acid, which is a substrate for several enzymes including the terminal allene oxide synthase (AOS) [14,15], while the 9-LOXs catalyze the synthesis of plant defense-related 9(S)-hydroperoxy-octadecatrienoic acid [16]. The subfamily of 9-LOX proteins is relatively more similar (~60%) than the 13-LOXs protein subfamily (~35%) [17].

LOX enzymes are encoded by a multi gene family and are intrinsic to plant growth and development [18]. Multiple genes have been reported in various plant species including six in Arabidopsis; 14 each in potato, tomato and rice; 15 in medicago; 18 in grape; 19 in soyabean; 20 in poplar; 23 in apple and cucumber and 122 in six legumes [19,20,21,22,23,24,25,26]. The 13-LOX derived oxylipins including JA and MeJA have been characterized for their roles during plant development and in response to biotic and abiotic stresses [27].

Increased LOX activity has been correlated with salt tolerance in tomato seedlings [28]. *CaLOX1* was reported to modulate abiotic stress responses via rapid scavenging of ROS and activation of defense-related marker genes in pepper [29]. The persimmon 9-LOX gene *DkLOX3* was found to enhance tolerance to abiotic stress but also promoted senescence [30]. Tomato *SlLOX4/D* was linked to jasmonate biosynthesis during plant development and defense [31] as well as to wound-induced stress [26]. It is also known that *SlLOX4/D* mutant, *spr8* (*suppressor of prosystemin-mediated responses8*), had jasmonate-dependent immune deficiencies ranging from the suppression of wound-responsive genes and severely compromised resistance to *Helicoverpa armigera* and *Botrytis cinerea* [32]. Jasmonate is known to induce the PDF1.2 transcription factor in Arabidopsis [33] and heat shock-regulated sesquiterpene synthesis in *Aquilaria sinensis* [34]. *AtLOX2* [35] and *AtLOX6* [36] transcripts have been linked to long-distance signaling in plants. It is also accepted now that LOX activity is a biological marker in plant stress tolerance [37].

We previously identified 14 *LOX* genes in the tomato genome [26]. Here, we delineate specific transcriptional responses of these *LOX* gene members in tomato leaves independently exposed to cold, heat, drought or salinity, stresses that impede plant growth. These findings link and further validate their usefulness as biological markers in stress tolerance/response/mitigation.

## 2. Materials and Methods

### 2.1. Plant Material

Wild type tomato (*Solanum lycopersicum* cv. Ailsa Craig) plants were grown in a temperature-controlled greenhouse under natural light conditions.

### 2.2. Abiotic Stress Treatments and Sample Collection

Four weeks old (28 days after transplanting) tomato plants were independently given a known stress as follows. Heat treatment was given in a growth chamber (EGC Corp., Chagrin Falls, OH, USA) with day/night (16/8 h) photoperiod, temperature 42 °C ± 0.15 °C and relative humidity (RH) of 50–70%. Leaf samples were then collected at 0, 0.5, 1, 6, 12 and 24 h. Cold treatment (4 °C) was given to whole tomato plants in a walk-in cold room at 200 μmol/m^2^ light and 16/8 h photoperiod. Leaf samples were then collected at 0, 1, 6, 12 and 24 h. Salt and drought treatments were given in a temperature-controlled greenhouse under natural light. Salt treatment involved irrigation of plants with 200 mL of 200 mM NaCl solution in a container containing metromix 360 soil (Sun Gro Horticulture, Agawam, MA, USA). Leaf samples were then collected at 0, 2, 6, 48 and 96 h. Drought treatment involved withholding watering of plants for seven days (168 h). Leaf samples were collected at 0, 24, 48, 72 and 168 h during which time watering was withheld. All the harvested samples were immediately frozen in liquid nitrogen and stored at −70 °C until used [38]. A minimum of three biological replicates were used for each time point where each biological replicate comprised of two technical replicates (*n* = 6). For each stress treatment, the 3rd leaf from the top was collected to maintain consistency, and three biological replicates were used. All experiments were repeated twice.

### 2.3. Mining the S. lycopersicum LOX Transcriptome

Transcriptome data for domesticated tomato (*S. lycopersicum* cv. Heinz) for the *LOX* gene family leaf tissue was extracted from the International Tomato Genome Sequencing Consortium (SGN; solgenomics.net) database (version ITAG 2.4). Reads per kilobase of transcript per million (RPKM) mapped reads values were retrieved, analyzed and are presented as heat maps. Heat maps were generated as previously described (https://discover.nci.nih.gov/cimminer/) [26,39].

### 2.4. Total RNA Extraction, Complementary DNA Synthesis and Quantitative RT-PCR

Frozen tomato leaf samples were ground with liquid nitrogen in a pestle and mortar to a fine powder. Total RNA was extracted from 100 mg powdered tissue using Plant RNeasy kit as per the manufacturer’s instructions (Qiagen, Germantown, MD, USA). Isolated RNA was treated with RNase-Free DNase (Qiagen) and cleaned using an RNeasy Mini Kit (Qiagen). RNA samples with an *A*_260/280_ ratio of 1.8–2 were subjected to agarose gel electrophoresis to ensure the presence of intact ribosomal RNA (rRNA) band ratios and selected for complementary DNA (cDNA) preparation [40]. A total of 2 µg RNA was used for cDNA synthesis using iScript Advanced cDNA synthesis kit (Bio-Rad, Hercules, CA, USA). Complementary DNA was diluted 10-fold for further use. Quantitative real-time PCR (qRT-PCR) was performed using SsoAdvanced Universal SYBR Green Supermix (Bio-Rad) in a Bio-Rad cycler (CFX96 Bio-Rad Real Time PCR machine). PCR conditions applied were: 95 °C for 10 min, 95 °C for 15 sec and 60 °C for 60 s (40 cycles), followed by a melt curve analysis [39]. Relative C_t_ values of gene expression were quantified using the ∆∆C_T_ method [41]. Reference genes, *SlTIP41* and *SlUBI3*, were used to normalize the expression of the target genes, and final data were calculated based on the geometric means of both [42,43]. Relative C_t_/C_q_ cycle was calculated by the Bio-Rad CFX Manager 3.1 based on the MIQE (minimum information for the publication of quantitative real-time PCR experiments) guidelines [44]. Relative fold changes were calculated as previously described [26,39]. Primer sequences used for qRT-PCR are listed in Appendix A with their SGN identity numbers. Relative qRT-PCR data represent average ± standard deviation from a minimum of three independent biological replicates for each gene.

### 2.5. Prediction of Abiotic Stress Related Cis-Regulatory Elements in SlLOX Promoters

For identification of cis-regulatory elements in the tomato *LOX* genes, proximal promoter sequences (–)2 Kb upstream to ATG for each *LOX* gene, were extracted from SGN. The upstream sequences were analyzed using the PlantCARE promoter analysis tool [45].

### 2.6. Data Analysis

XLSTAT suite was used for statistical analysis. A one-way ANOVA was performed using a Tukey (HSD) test. Significant differences in the data from abiotic stress experiments were calculated against non-treated control (0 h) samples and were categorized at * *p* < 0.05, ** *p* < 0.01 and *** *p* < 0.001 for each analysis as described previously [39,46].

## 3. Results

### 3.1. Lipoxygenases Catalyze Synthesis of Oxylipin Signaling Molecules

Plant lipoxygenases (LOXs) classify into 9-LOX and 13-LOX enzymes involved in the biosynthesis of oxylipins, including jasmonic acid and its derivatives. The 9-LOX enzyme catalyzes the conversion of 18:2 linoleic acid (LA) and 18:3 linolenic acid, respectively, to 9-hydroperoxide octadecadi(tri)enoic acids (9-HPOD/T) while the 13-LOX enzyme catalyzes the synthesis of 13-hydroperoxide octadecatrienoic acid (13-HPOT) from linoleic/linolenic acids (Figure 1). Allene oxide synthase (AOS) converts 9-HPOD/T to 9,10-epoxy octadecadienoic acid (9,10-EOD), which is followed by formation of either 10-OPDA (oxo-phytodienoic acid) or ketols (Figure 1). The 13-LOX enzyme catalyzes the formation of 13-HOPT, which generate substrate for the synthesis of jasmonic acid [47,48]. We previously identified 9-LOX and 13-LOX sub-families of tomato based on the catalysis of substrate carbon oxidation position and their likeness with previously characterized plant LOXs [26]. Tomato 9-LOX sub-family constitutes nine genes, namely, *SlLOX1, 2, 5, 6, 7, 8, 9, 13* and *14* and the 13-LOX sub-family is comprised of five genes, namely, *SlLOX3, 4, 10, 11* and *12*. The predicted open reading frames for the members of LOX family range from 2526 to 2736 nucleotides [26].

### 3.2. LOX Sub-Family Gene Expression in the Domesticated Tomato (S. lycopersicum cv. Heinz)

Expression patterns of *LOX* gene family members in leaf tissue were created by analyzing the available leaf transcriptome data for the *SlLOX* gene family for *S. lycopersicum* (SGN). Both the 9-LOX and 13-LOX sub-family members were differentially expressed in tomato leaf tissues (Figure 2) [26]. Among the 9-LOX subfamily genes, *SlLOX6* was the highly expressed gene and *SlLOX1, 7,* 8 and 14 were the least expressed in the leaf tissue. *SlLOX13* was not expressed. Among the 13-LOX subfamily members, *SlLOX3*, *4, 11* and *12* were highly expressed whereas *SlLOX10* was least expressed.

### 3.3. Differential Expression Kinetics of 9- and 13-LOX Genes in S. lycopersicum cv. Ailsa Craig in Response to Different Abiotic Stresses

#### 3.3.1. Heat Stress

Time dependent qPCR-based expression patterns of 14 *LOX* genes in tomato leaves exposed to 42 °C for up to 24 h are shown in Figure 3. An immediate downregulation in the expression of five *9-LOX* family genes (*SlLOX1*, *2, 7, 8* and *14*) and two 13-LOX family members (*SlLOX3* and *10*) within the first 30 min of heat exposure were apparent (Figure 3; Appendix A). Of these genes, downregulation remained imminent until 6 h of heat stress for *SlLOX1*, *SlLOX8* and *SlLOX14* genes (9-LOX family). It was also noted that the relative expression of *SlLOX9* (9-LOX family) and *SlLOX3* and *SlLOX10* (13-LOX family members) genes remained low during the 24 h exposure to heat stress (Figure 3). In contrast, higher expression of 9-LOX family members, viz., *SlLOXs* 5, 6, 7 and of 13-LOX family members, namely, *SlLOX4*, *SlLOX11* and *SlLOX12*, was evident, the expression remaining significantly high throughout the 24 h exposure except for *SlLOX11* (Figure 3). Other *SlLOX* gene members that were upregulated at 24 h stress included 9-LOX members *SlLOXs1, 2, 5, 7* and *14* together with that of 13-LOX member *SlLOX4*. Thus, tomato *LOX* gene family members selectively and differentially respond to heat stress.

#### 3.3.2. Cold Stress

Cold stress of tomato leaves was tested at 4 °C for a duration of 24 h. Time dependent qPCR-based expression patterns of 14 *LOX* genes in tomato leaves exposed to 4 °C are shown as bar (Figure 4) and line (Appendix A) graphs. Like the response patterns to heat stress, tomato leaf *LOX* gene family members differentially responded to cold stress. Expression of 9-LOX *SlLOX2* and 13-LOX *SlLOX10* gene members was simultaneously downregulated at 1 h and remained downregulated throughout the exposure period of 24 h (Figure 4; Appendix A). The earliest transcript upregulation response, within 1 h of cold exposure, was apparent for four *LOX-9* family gene members (*SlLOX1, 6, 13* and *14*) and three of the *LOX-13* family (*SlLOX3*, *4* and *12*) members (Figure 4). A number of the *9-LOX* gene family members, viz., *SlLOX7, 8, 13* and *14*, as well as the *13-LOX* member *SlLOX12* registered peak activation at 6 h of cold exposure and were downregulated thereafter; however, their gene expression was still at a higher level compared to that at 1 h of cold stress. The *9-LOX* gene members *SlLOX5* and *SlLOX14* as well as *13-LOX SlLOX4* maintained a high level of expression at 24 h of cold exposure. As seen for their response to heat stress above, tomato *LOX* gene family members also responded differentially to cold stress.

#### 3.3.3. Drought Stress

Drought stress led to the downregulation of four 9-LOX family members—*SlLOX1*, *SlLOX5*, *SlLOX7* and *SlLOX14* as well as of two 13-LOX members *SlLOX4* and *SlLOX10* within 24 h of water retention (bar graph Figure 5; line graph Appendix A). At 48 h of drought, the expression of seven out of nine 9-LOX family members *SlLOX1, 2, 6, 7, 8, 9* and *13* and three of the five 13-LOX members *SlLOX3, SlLOX10* and *SlLOX11* were upregulated. Notably, by 72 h of drought stress, eight *SlLOXs* 1, 2, 6, 7, 8, 9, 13 and 14 (9-LOX family) were highly upregulated than the 0 h control and similar trend was observed for the expression of four of the five 13-LOXs *SlLOX3*, 10, 11 and 12 (Figure 5). By 168 h of drought stress, most of the *SlLOX* members continued to be expressed at a higher level except for *SlLOX2* and *SlLOX8* genes. *SlLOX5* of the 9-LOX family remained suppressed throughout the drought period.

#### 3.3.4. Salt Stress

Salt stress led to the downregulation of six members of 9-LOX family, viz., *SlLOX2*, *SlLOX5*, *SlLOX7*, *SlLOX8*, *SlLOX13* and *SlLOX14* and only *SlLOX12* of the 13-LOX family within 2 h of treatment (Figure 6). Among these, those members that continued to be further downregulated at 48 or 96 h of salt treatment include four *LOX* genes—*SlLOX5*, *SlLOX8*, *SlLOX14* and *SlLOX12*. The peak upregulation was evident in a number of *LOX* genes, viz., *SlLOX1* (48 h), *SlLOX2* (48 h), *SlLOX6* (6 h), *SlLOX7* (48 h, 96 h), *SlLOX9* (48 h), *SlLOX3* (48–96 h), *SlLOX10* (48–96 h) and *SlLOX11* (6 h, 96 h) while *SlLOX4* was non-responsive to salt (Figure 6; Appendix A).

### 3.4. Abiotic Stresses-Related Cis-Elements that are Predominant in the SlLOX Promoters

We next looked for the presence of known stress-specific cis-elements in the *LOX* family of genes (Table 1). Notably, heat response elements (HSE) were found present in the promoters of both *9-LOX* (*SlLOX2, 5, 6, 9* and 13) and *13-LOX* (*SlLOX3, 4, 10* and *12*) sub-family gene members. Low temperature response elements (LTR) were present in the promoters of three *LOX* genes, *SlLOX5* and *SlLOX7* (9-LOX sub-family) and *SlLOX10* (13-LOX sub-family). Like the HSE element, drought related*-*MYB binding site (MBS) was predominant in the promoters of nine *LOX* genes, *SlLOX5, 7, 8, 9, 13* and *14* (9-LOX subfamily) and *SlLOX3, 4* and *10* (13-LOX sub-family). MYBHv1-related binding site (CCAAT-box, a stress and defense inducible element) was found on the promoter of *SlLOX9* (9-LOX sub-family) and *SlLOX13* (13-LOX subfamily).

## 4. Discussion

Lipoxygenases (LOXs—9-LOX and 13-LOX) catalyze synthesis of important signaling molecules such as jasmonates (jasmonic acid and methyl jasmonate) as well as oxylipins, with significant role(s) in plants during development and in response to abiotic and biotic stresses [8,10,29]. Tomato is susceptible to most abiotic stresses including heat, cold, drought and salinity, each of which can impact tomato yield. Environmental stress-based responses of LOX have been reported previously but they have not been compared under similar isogenic background [5,9,10]. We demonstrated here distinct and differential responses of specific gene members of 9-LOX and 13-LOX sub-families in tomato to four separately imposed abiotic stressors and highlight the distinct regulation of *LOX* gene family members in plant responses to stress. Such information is novel and important for developing molecular markers associated with adverse environmental conditions in order to help breeding for cultivars with higher stress resilience and thereby higher crop productivity.

A collation of the time-dependent responses of single and multiple *LOX* family of genes specific to each abiotic stress is presented in Figure 7. Three groups emerged distinctly: One included *LOX* gene members that were downregulated soon after a particular abiotic stress was imposed (immediate downregulation) and included gene members that were unique to each stress; the second group included *LOX* gene members that immediately upregulated following a particular stress (immediate upregulation) and included gene members that were unique to heat, salt or drought stress and the third group was made of single gene members, which were non-responsive to either heat (*LOX9*), cold (*LOX9*) or salt (*LOX4*) stress (Figure 7). It was also evident that all the *LOX* genes responded positively or negatively to drought. Thus, these data delineated the positive or negative regulation of each *SlLOX* family member with imposed stress. This novel information on *LOX* gene family members should be helpful in future studies in developing strategies to combat plant stress, generating stress-resistant transgenic plants and/or searching for unique tomato germplasm rich in a particular *LOX* gene family member.

Most of the heat stress-upregulated tomato *LOX* genes analyzed were found to harbor cis-element HSE in their promoters indicating that related transcription factors might interact and induce their expression. However, other members *SlLOX1, 7, 8, 12* and *14* did not harbor HSE cis-element in their promoters and yet were induced during heat stress except for the *SlLOX8* gene, suggesting that some other unknown factors may facilitate their upregulation during heat stress. Similarly, low temperature responsive motif (LTR, one copy of CCGAAA) was found only in the promoters of two genes *SlLOX5* and *SlLOX7,* even though many other members were regulated by cold stress. This suggests that a majority of the cold-regulated *LOX* genes in tomatoes were induced via some other yet unknown transcription factors and/or regulatory genes. Cold exposure is known to increase endogenous JA, for instance, in rice seedlings, which was correlated with the expression of *OsLOX2* [49]. Similarly, exogenous MeJA treatment was found to improve freezing tolerance in *Arabidopsis* seedlings while inhibiting JA biosynthesis and signaling pathway made the seedlings hypersensitive to freezing stress [50].

Any association of previously studied cis-elements in relation to salt stress was not found in our studies. The CCAAT box has been associated with nuclear binding transcription factors that play a role in abiotic stresses, such as drought [51,52,53,54], salinity [55,56] and cold [56,57]. Interestingly, some of the LOX gene promoters analyzed in this study were rich in CCAAT box. It is possible that tomato LOX proteins activated in response to salt stress may be a target of these transcription factors and regulatory genes, which utilize CCAAT boxes. In pepper, *9-LOX* gene *CaLOX1* has been reported to play a role in high salinity and drought tolerance [29] while drought is known to induce *PgLOX3* expression in *Panax ginseng* [9]. Here, a number of *SlLOX* genes were found to differentially respond to drought. Based on the strength of upregulation in response to drought, *SlLOXs3*, *6* and *8* genes were among the highly expressed LOXs in response to drought. Interestingly, *SlLOX*3 promoter was found decorated with the MYB binding site (MBS), which may enable its drought inducibility. It is also noted that drought stress activated *LOX* genes of both 9-LOX and 13-LOX family members.

Salt stress is known to activate tomato *LOX* genes [5]. Pretreatment of *Pisum sativum* seedlings with MeJA was also found to counter the salt stress [58] Thus, a role of MeJA/JA as well as related metabolites seems to be important for protection against salt stress in plants. The data presented here are in tune with these previous studies and bring to fore differential response of tomato *SlLOX* genes to high salt irrigation.

## 5. Conclusions

This study identified members of the *9-LOX* and *13-LOX* family of genes in tomato and demonstrated differential and specific expression of each LOX family member in response to four different abiotic stresses. These findings linked and further validated their usefulness as biological markers in stress tolerance/response/mitigation. Three distinct groups emerged: Tomato *LOX* gene members that were downregulated immediately in response to an abiotic stress and gene members unique to each stress; *LOX* gene members that were upregulated in response to a particular stress and gene members unique to heat, salt or drought stress and a group of single gene members, which were non-responsive to either heat (*SlLOX9*), cold (*SlLOX9*) or salt (*SlLOX4*) stress. The data presented here are useful and provide a firm background for future investigations in search of robust and novel *cis*-acting elements differentially present in the promoter regions of the LOX family members in plants. Together, their genetic manipulation should help in devising strategies for improving plant tolerance to different abiotic stresses.

## Figures and Tables

**Figure 1 genes-10-00683-f001:**
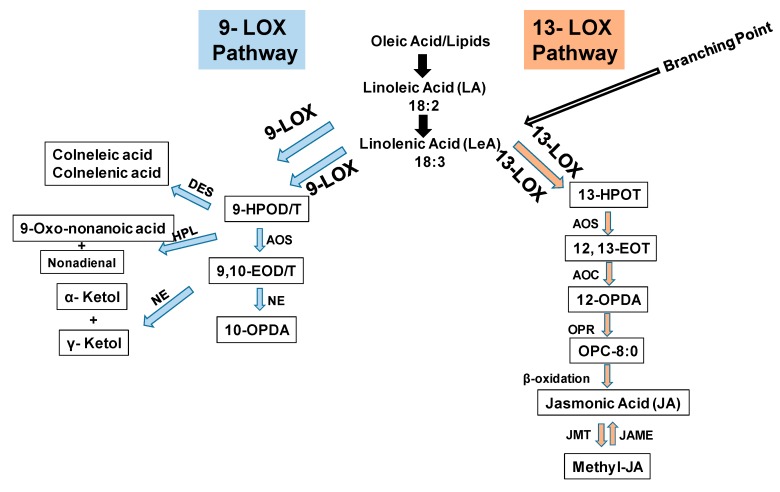
A schematic representation of 9-LOX and 13-LOX pathways in plants [47,48]. Abbreviations: HPOD: 9 or 13-hydroperoxide linolenic acid; 13(S) or 9(S)-hydroperoxylinolenic acid; OPDA: 12-oxo-phytodienoic acid; LOX: Lipoxygenase; AOS: Allene oxide synthase; AOC: Allene oxide cyclase; OPR: Oxo-phytodienoic acid reductase; JMT: Jasmonic acid carboxyl methyltransferase; JAME: Methyl jasmonate esterase; HPL: Hydroperoxide lyase; DES: Divinyl ether synthase. NE: Non-enzymatic.

**Figure 2 genes-10-00683-f002:**
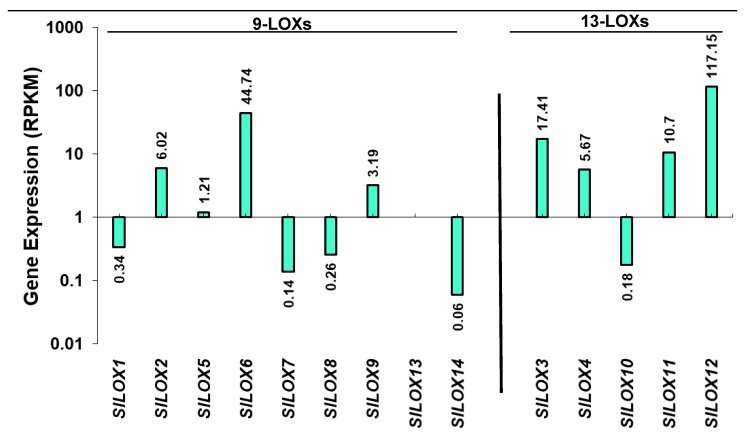
Relative transcript abundance of *9-LOX and 13-LOX* gene family members in the leaf tissue of *S. lycopersicum* cv. Heinz tomato. The bars represent processed reads per kilobase of transcripts per million mapped reads (RPKM) for the tomato leaf *LOX* genes derived from RNA-seq data (International Tomato Genome Sequencing Consortium [SGN] database) as described in the Materials and Methods section [26]. Y-axis as Log2 scale with inverted axis shows gene members minimally active in the leaf.

**Figure 3 genes-10-00683-f003:**
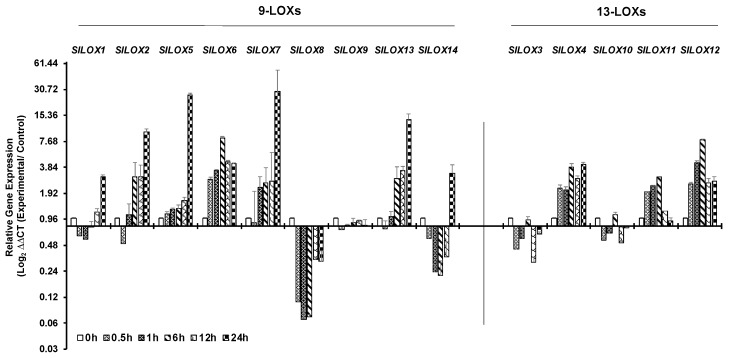
qRT-PCR analysis of tomato (*S. lycopersicum* cv. Ailsa Craig) *LOX* gene family members in response to heat stress. Graphical representation qRT-PCR data of tomato 9-*LOX* genes and *13-LOX* genes in response to heat treatment at 42 °C for up to 24 h. Non-treated control (0 h) was used as a calibrator in the qRT-PCR data calculation and for defining statistical significance of treatment data points at levels * *p* < 0.05, ** *p* < 0.01 and *** *p* < 0.001. Details are given in Appendix A. A minimum of three biological replicates, where each biological replicate was comprised of two technical replicates, were used for each time point. Two housekeeping genes, *SlTIP41* and *SlUBI3*, were used to normalize the expression of target genes.

**Figure 4 genes-10-00683-f004:**
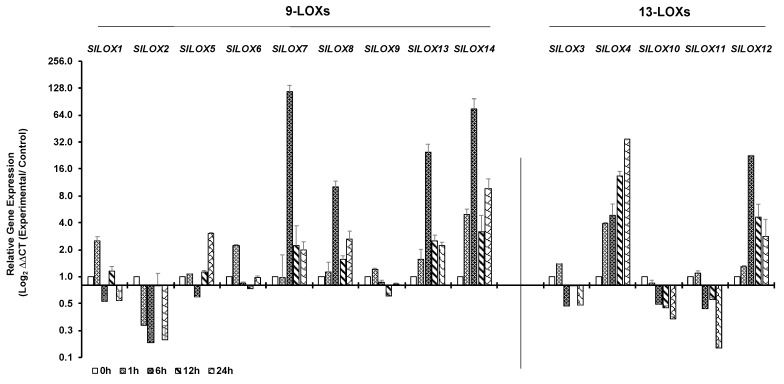
qRT-PCR analysis of tomato (*S. lycopersicum* cv. Ailsa Craig) *LOX* gene family members in response to cold stress. Graphical representation of qRT-PCR data of tomato 9-*LOX* genes and *13-LOX* genes in response to cold treatment at 4 °C for up to 24 h. A non-treated control (0 h) was used as calibrator in qRT-PCR data calculation and for defining statistical significance of treatment data points at levels * *p* < 0.05, ** *p* < 0.01 and *** *p* < 0.001. Details are given in Appendix A. A minimum of three biological replicates, where each biological replicate was comprised of two technical replicates, were used for each time point. Housekeeping genes, *SlTIP41* and *SlUBI3*, were used to normalize the expression of target genes.

**Figure 5 genes-10-00683-f005:**
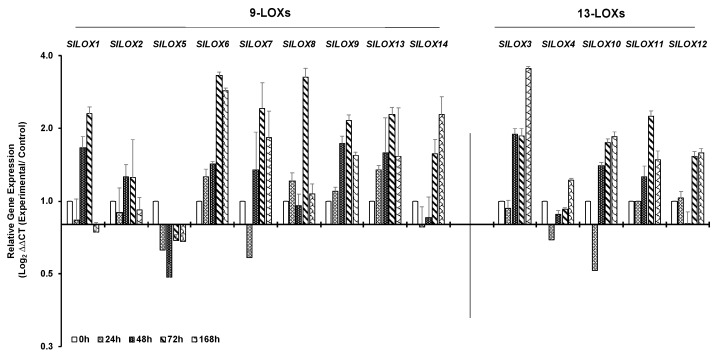
qRT-PCR analysis of tomato (*S. lycopersicum* cv. Ailsa Craig) *LOX* gene family members in response to drought stress. Graphical representation of 9-*LOX* genes and *13-LOX* genes in response to drought (water withheld for 168 h). A non-treated control (0 h) was used as a calibrator in qRT-PCR data calculation and for defining statistical significance of treatment data points at levels * *p* < 0.05, ** *p* < 0.01 and *** *p* < 0.001. Details are given in Appendix A. A minimum of three biological replicates, where each biological replicate was comprised of two technical replicates, were used for each time point. *SlTIP41* and *SlUBI3* housekeeping genes were used to normalize the expression of target genes.

**Figure 6 genes-10-00683-f006:**
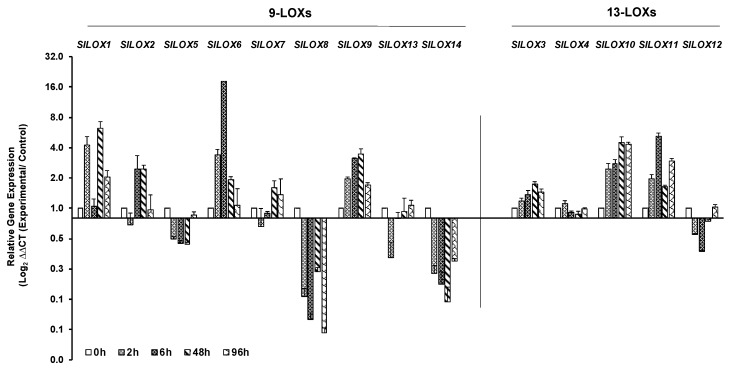
qRT-PCR analysis of tomato (*S. lycopersicum* cv. Ailsa Craig) *LOX* gene family members in response to salt stress. Graphical representation of qRT-PCR data of tomato 9-*LOX* genes and *13-LOX* genes in response to salt stress: 200 mL of 200 mM salt solution was applied daily to plants for 96 h. Non-treated control (0 h) was used as calibrator in qRT-PCR data calculation and for defining statistical significance of treatment data points at levels * *p* < 0.05, ** *p* < 0.01 and *** *p* < 0.001. Details are given in Appendix A. A minimum of three biological replicates, where each biological replicate was comprised of two technical replicates, were used for each time point. Housekeeping genes, *SlTIP41* and *SlUBI3*, were used to normalize the expression of target genes.

**Figure 7 genes-10-00683-f007:**
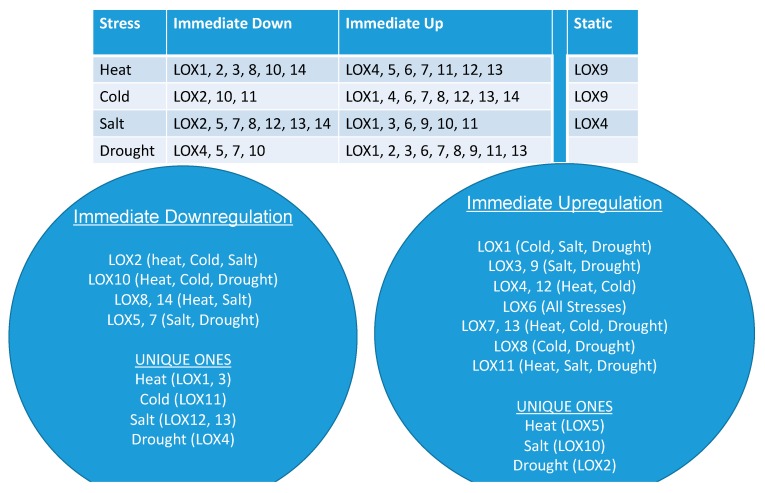
Summary of the immediately downregulated, immediately upregulated and non-responsive (static) tomato *9-LOXs* and *13-LOXs* genes in response to the four different and independently applied abiotic stresses.

**Table 1 genes-10-00683-t001:** Identification of the abiotic stress response cis-elements in tomato (*Solanum lycopersicum* L.) 9-LOX and 13 LOX sub-family genes used in this study.

Sub-Family	Gene Name	Heat	Cold	Drought	Salt	Others
9-LOX	*SlLOX1*	?	?	?	?	ARE-box
	*SlLOX2*	HSE	?	?	?	TC-rich repeats, AT-Rich sequence, ARE-box
	*SlLOX5*	HSE	LTR	MBS	?	TC-rich repeats
	*SlLOX6*	HSE,	-		?	ARE
	*SlLOX7*	?	LTR	MBS	?	TC-rich repeats, Box-W1, ARE
	*SlLOX8*	?	?	MBS	?	TC-rich repeats, Box-W1, ARE
	*SlLOX9*	HSE	CCAAT-BOX	MBS, CCAAT-BOX	CCAAT-BOX	ARE
	*SlLOX13*	HSE	CCAAT-BOX	MBS, CCAAT-BOX	CCAAT-BOX	TC-rich repeats, ARE
	*SlLOX14*	?	?	MBS	?	TC-rich repeats
13-LOX	*SlLOX3*	HSE	CCAAT-BOX	CCAAT-BOX	CCAAT-BOX	ARE, MBS, TC-rich repeats
	*SlLOX4*	HSE	?	MBS	?	TC-rich repeats, ARE
	*SlLOX10*	HSE	LTR	MBS	?	TC-rich repeats, Box-W1, ARE
	*SlLOX11*	?	?	?	?	
	*SlLOX12*	HSE	?	?	?	

Question mark (?) indicates that no element associated with a certain response was found in the indicated *LOX* gene promoter.

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
