# Peer review of "Transcript Abundance Patterns of 9- and 13-Lipoxygenase Subfamily Gene Members in Response to Abiotic Stresses (Heat, Cold, Drought or Salt) in Tomato (Solanum lycopersicum L.) Highlights Member-Specific Dynamics Relevant to Each Stress"

_genes, 2019, doi:10.3390/genes10090683_

Round 1
Reviewer 1 Report
The authors showed the expression of LOX genes in Tomato in different stress conditions. It is interesting to see the different expression patterns of LOX genes; however, I have several comments:
1) The work is difficult to read and needs an extensive English editing mainly on the abstract and the introduction sections.
2) Table 1 has the same information as Table 1 of your article published in Journal of Plant Physiology.
3) Figure 2 has the same information of Figure 4 of your article published in Journal of Plant Physiology.
4) What is the purpose of putting the same information in two different formats (supplementary figures 1-4 and the main figures 3-6)?
5) Although it is very interesting to see all these expression data, the manuscript is very descriptive and needs integration and discussion.
For example:
Can you compare the salt treatment with the drought treatment?, why is so different the expression patterns of the different genes between these two stress conditions?, what is the osmotic potential of the soil during the salt and the drought treatments? Why are these genes (SlLOX8 and SlLOX14) downregulated in heat and cold stress conditions?, how can you explain that?
6) It is interesting to see all the abiotic stress response cis-elements in the promoter regions of these genes. Do you considered all the variants of each element? or are there other elements in the target genes that responds to a specific stress conditions? or is there a possibility to find elements of transcript regulation in introns sequences?.
It is difficult to explain why many genes regulated during the different stress conditions do not have the elements that are needed to respond to those stress conditions: for example, SlLOX8 is highly regulated by heat stress and do not have HSE elements-
Minor Essential Revisions
1) In the title you have “specific dynamics specific to each stress”.
2) Lane 32. “Decorated” sounds strange.
3) Lane 43 after salinity put (-).
4) Line 49 after jasmonate put (JA)
5) Line 70 insert a space after “The”
Reviewer 2 Report
Manuscript by Upadhyay et al. describes transcriptional regulation of the LOX gene family genes in tomato. The authors identified and characterized tomato gene family members in an earlier publication. Here in the present work, they present results on the comprehensive characterization of the expression profile of these genes under heat, cold stress, salt and drought stress at different time points of the imposition of the signals. Thus, the present study, in general provides some useful insights into possible functional roles of specific family members in abiotic stress response of tomato. In addition, authors predicted possible role of cis- acting elements present in the promoter regions in with respect to the stress responsive expression of these genes. In general the manuscript is well written and experiments have been elegantly planned along with nice representation of the data. I believe that the present study should be useful as a firm background in future ventures focused on the functional characterization on LOX family members.
Author Response
Please see the attachment

This manuscript is a resubmission of an earlier submission. The following is a list of the peer review reports and author responses from that submission. The previous reviewers did not agree to review again
Round 1
Reviewer 1 Report
The manuscript Upadhyay et al describes expression analysis of LOX gene family in wild tomato relative and a domesticated tomato cultivar. The technically, experiments were conducted properly, and the authors detected differential expression of LOX isoforms during various abiotic stresses. There are few points that authors need to address.
Major point
1. The authors use Solanum pimpinellifolium and Solanum lycopersicum cv. Ailsa Craig as their materials. The authors claimed that the transcript level differences between two species are due to domestication. This reviewer thinks that there are insufficient data to conclude if the observed difference between wild and domesticated tomato samples truly represent domestication process or simple variability among domesticated species. The authors would need to check multiple tomato cultivars to imply this.
Minor points
Abstract and page 6, The authors discuss gain or loss of LOX genes/transcripts. These statements are based on RT-qPCR experiments. Then, only gain or loss of LOX "expression" can be implied.
Abstract last sentence " LOX genes in the more recent S. lycopersicum cultivar appear to have
positively multiplied during domestication process; however, SlLOX13 was lost." This sentence does not make sense. The authors did not analyze gene copy numbers or genome structures in this study.
Reviewer 2 Report
This study used data mining of public databases and qRT-PCR to investigate patterns of expression of members of two lipoxygenase gene families in cultivated tomato and its closest wild relative. Not surprisingly, individual genes in these families had different levels of expression in each of these species. However, these differences in gene expression are severely misinterpreted in the text. For instance, the last two sentences in the abstract say that these different levels of transcripts "... shed some light on the the gain or loss of LOX genes during tomato domestication. LOX genes in the more recent S. lycopersicum cultivar appear to have positively multiplied during domestication process; however SlLOX13 was lost."
None of this is true. Transcript levels in one tissue measured at one time cannot say anything at all about the gain or loss of genes. SlLOX13 certainly was not lost in S. lycopersicum, as data presented in figures 3,4,5,and 6 all show SlLOX13 transcripts in S. lycopersicum.
In lines 180 through 185, the authors state that common transcriptional levels of a gene in both S. pimpinellifolium and S. lycopersicum indicate conservation of function. There is no biological basis for this claim at all.
On line 196, the authors claim that increase transcript abundance reflects a gain in function for the gene. There is no legitimate way to infer a gain of function from transcript level, otherwise genes would be gaining new functions every time some environmental or physiological cue altered their expression level.
In lines 201 and 202, the authors claim that transcript levels "revealed a gain of both subfamily of of genes during domestication..." It is not clear what is being gained here.
When I first read the abstract, I was excited about the potential loss of LOX13 during tomato domestication, and I was disappointed when I saw that it was expressed in figures 3-6. I assumed that this discrepancy was due to an unfortunate misunderstand of English. However, repeated statements about the gain and loss of genes and the gain and loss of function based merely on transcript levels leads me to believe that at least one of the authors lacks sufficient understanding of the biology involved here, and the other co-authors have not bothered to correct this.
The data presented probably are sound, and it is helpful to know which genes respond to which stress, but their evolutionary interpretation is completely wrong and misguided.
If the authors really wanted to compare expression of LOX genes between two species, they should have performed the qRT-PCR experiments under various stress conditions on both species, not just S. lycopersicum.
Minor points -
Figure 2 - It is not clear what the authors mean by 'transcript occupancy.' The only way I have seen this term being used is in ribosome profiling where mRNAs are bound to the ribosome. Also it is not clear why the scale for 'transcript occupancy' changes between 2B and 2C. What are the units for the Y axis in 2C?
For Figures 3, 4, 5, and 6, the graphs of 9-LOX expression and 13-LOX expression should be labeled 'A' and 'B', respectively. Also, the legend refers to levels of P-values indicated by asterisks, but there are no asterisks on the graphs.
Reviewer 3 Report
The manuscript by Upadhyay et al. tries to address member-specific differences in the response of tomato lipoxygenase genes against several abiotic stresses. Although the issue has a good potential, the experimental part is insufficient to conclude something relevant.
- The comparison in the expression of lipoxygenase genes between wild and cultivated tomato does not provide any information if it is not conveniently checked by experiments in control and abiotic-stressed leaves.
- Expression analyses seem preliminary results. Member-specificities should be corrrelated with the quantification of final products in both 9-LOX and 13-LOX pathways. This kind of experiments would permit to conclude the actual relevance of each LOX member in the plant responses to the treatments.
- The relevance of putative binding sites for transcription factors in promoters should be experimentally tested. Many sites are recognized by multiple transcription factors, which could be associated to one or many different stresses. Besides, unknown binding sites for stress-associated transcription factors will be discover, which would likely change the reported assignment of putative binding sites to specific gene-promoters.
In addition, I have found some huge mistakes. For example, in the abstract section, the final sentence states that cultivated tomate has gained several genes during tomato domestication. Likewise, PDF1.2 is considered a transcription factor in the introduction section.
Finally, in a quick Blast on tomato genome 3-4 additional LOX genes seem be present in both wild and cultivated tomato.
Author Response
Reviewer 1
RESPONSE: We thank this reviewer for comments. We have made a major revision of our previous
version. We hope this reviewer is satisfied with the revision.
Reviewer 2
RESPONSE: We apologize for misinterpretation of the data and thank the reviewer for an objective
review. In the light of comments from all three reviewers, we have deleted the domestication part
(old Figure 3 and related material) from the revised version. We have substantially revised the
manuscript and focus is entirely on differential regulation and kinetics of 14 LOX genes in response
to 4 different abiotic stressors which provide novel information and distinguish between different
LOX gene members. We believe these findings are beneficial to plant science community not only
for the basic science but also consideration of future biotechnological manipulations.
Statistical differences in the supplementary data with line graphs for figures 3, 4, 5 and 6 are also
presented.
Reviewer 3
RESPONSE: We thank the reviewer for comments and in view of these we have revised the
manuscript. In the earlier version we presented available transcriptome data to give an insight and
establish a correlation between Solanum pimpinellifolium and Solanum lycopersicum for LOX
family. However, we recognized that our presentation arose confusion. Thus, we have deleted the
Solanum pimpinellifolium transcriptome data. From the revised version. The abiotic stress
response treatments shown are for a modern cultivar of Solanum lycopersicum. We have also
changed the abstract and removed the last sentences as suggested by this reviewer.
Additional Remarks: We hope the reviewers’ will see that we have taken time to understand their
critique and accordingly revised the original version of this manuscript. We hope that they will
appreciate our contention that this study will be of importance to other researchers interested in
knowing that different members of the same gene family respond specifically and differentially to
different abiotic stressors.